# Analysis of Publications on Health Information Management Using the Science Mapping Method: A Holistic Perspective

**DOI:** 10.3390/healthcare12030287

**Published:** 2024-01-23

**Authors:** Dilaver Tengilimoğlu, Fatih Orhan, Perihan Şenel Tekin, Mustafa Younis

**Affiliations:** 1School of Business, Department of Business, Atılım University, 06830 Ankara, Türkiye; dilaver.tengilimoglu@atilim.edu.tr; 2Gülhane Vocational School of Health, University of Health Sciences, 06010 Ankara, Türkiye; fatih.orhan@sbu.edu.tr; 3Vocational School of Health Services, Ankara University, 06290 Ankara, Türkiye; 4School of Public Health, Jackson State University, Jackson, MS 39213, USA; younis99@gmail.com

**Keywords:** health information management, electronic records, bibliometric analysis

## Abstract

Objective: In the age of digital transformation, there is a need for a sustainable information management vision in health. Understanding the accumulation of health information management (HIM) knowledge from the past to the present and building a new vision to meet this need reveals the importance of understanding the available scientific knowledge. With this research, it is aimed to examine the scientific documents of the last 40 years of HIM literature with a holistic approach using science mapping techniques and to guide future research. Methods: This study used a bibliometric analysis method for science mapping. Co-citation and co-occurrence document analyses were performed on 630 academic publications selected from the Web of Science core collection (WoSCC) database using the keyword “Health Information Management” and inclusion criteria. The analyses were performed using the R-based software Bibliometrix (Version 4.0; K-Synth Srl), Python (Version 3.12.1; The Python Software Foundation), and Microsoft^®^ Excel^®^ 2016. Results: Co-occurrence analyses revealed the themes of personal health records, clinical coding and data quality, and health information management. The HIM theme consisted of five subthemes: “electronic records”, “medical informatics”, “e-health and telemedicine”, “health education and awareness”, and “health information systems (HISs)”. As a result of the co-citation analysis, the prominent themes were technology acceptance, standardized clinical coding, the success of HISs, types of electronic records, people with HIM, health informatics used by consumers, e-health, e-mobile health technologies, and countries’ frameworks and standards for HISs. Conclusions: This comprehensive bibliometric study shows that structured information can be helpful in understanding research trends in HIM. This study identified critical issues in HIM, identified meaningful themes, and explained the topic from a holistic perspective for all health system actors and stakeholders who want to work in the field of HIM.

## 1. Introduction

The healthcare industry is rapidly evolving to meet the demands and opportunities of digital transformation [1,2,3]. This evolutionary process includes some headlines such as “technology adoption” [4], “standardized clinical coding” [5], “health information system success” [6], “electronic record types” [7], “personal health information management” [8], “consumer health informatics” [9], “eHealth and eMobile health technologies” [10], “country frameworks and standards for HIS” [11], and “critical health information management”. The acceptance of technology is the foundation of HIM [12]. How healthcare professionals and managers approach technology is critical to the successful functioning of HIM systems [13]. This acceptance process increases the effectiveness of clinical coding systems that provide a standardized language and methodology [14,15]. For example, international coding systems such as the International Classification of Diseases (ICD)-10 positively impact the effectiveness and efficiency of HISs [16].

The effectiveness of HISs requires accurate data collection, analysis, and exchange processes. This means that the standardization of clinical coding has a direct impact on the success of HISs [17,18]. Electronic health records and e-health technologies are essential for measuring and improving the success of systems. These platforms are used to improve the quality of patient care and ensure effective data management [19]. These systems, which allow individuals to manage their own health information with effective data management, are critical to health informatics for all health system stakeholders. The ability of individuals to manage their health information more effectively enables faster and more informed health decisions [20]. In this context, national and international standards and frameworks are of great importance to ensure efficiency and harmonization in HIM, integration between different topics, improvement of the overall quality of health services, increased efficiency, and cost control [11].

When it comes to HIM, many different terms and abbreviations are often encountered. Many different types of electronic records are kept during the delivery of healthcare services. Electronic medical records and electronic health records are the most important. Other types of records include personal health records, genomic point-of-care records, oncology patient records, and records for drug/pharmaceutical networks [21]. Of all the record types, the ones that have undergone the most research are electronic health records (EHRs), electronic medical records (EMRs), and personal health records (PHRs) [22]. Clearly defining these record types as terms is critical to establishing standards and more effectively implementing HIM [23].

These terms represent different functions, capabilities, and requirements, and there are clear differences between them. Clarity of definitions avoids confusion among all healthcare stakeholders, facilitates data exchange, and improves the overall quality of healthcare. In addition, these definitions are important for policymakers, as national and international standards and frameworks require an understanding of what these terms mean. In this context, a clear and precise definition of terms contributes to a better understanding of the different dimensions of HIM and to more effective work in this area. The text box in Appendix A defines the most important terms by compiling them from various sources.

HISs play a critical role in planning, monitoring, and evaluating health services. The effectiveness and efficient use of HISs can influence the quality of health services [24,25]. HIS components such as health information technology (HIT) and electronic health records (EHRs) have significant potential to improve the efficiency and quality of healthcare [26,27]. For example, Blumenthal and Tavenner (2010) [28] discussed the “meaningful use” regulation of EHRs, highlighting how this technology can add value for healthcare providers and patients.

HISs provide a critical infrastructure for the efficiency and effectiveness of healthcare services and facilitate communication and collaboration among healthcare providers, patients, and policymakers. The effective design and implementation of health information technology can improve the quality and reduce the cost of healthcare. Healthcare providers rely on HISs to organize and store patient information, plan treatment, and communicate effectively with patients. At the same time, HISs play an important role in areas such as patient safety and the prevention of medical errors [29]. Several studies have examined the impact of HIT on the quality and efficiency of healthcare services. Researchers such as Chaudhry et al. (2006) [25] and Buntin et al. (2011) [24] have shown that HIT can positively affect the quality, efficiency, and cost of healthcare services. In addition to these benefits of HIM, it is important to consider the costs of installing these systems, Internet usage, data security, and training staff to use these systems.

According to the HIM literature, there is an evolution in the process of resource management and adaptation to technological change [30]. Although previous reviews have traced these developments [30], it is clear that there is a need for a more comprehensive analysis of this rapidly evolving area of research, whose contribution to the health sector is increasing in parallel with the use of technology. Based on this gap, the aim of this study is to conduct a bibliometric analysis of publications on HIM using science mapping techniques. Within this framework, this study seeks to answer the following three main questions:

1. What are the most prolific authors, most cited journals and articles, and most prominent institutions, countries, and trending topics in HIM?

2. What are the most critical studies in HIM? What are the dynamics in the development of the intellectual structure of the field?

3. Can the conceptual structure of HIM be identified?

It is expected that the research will make two major contributions to the literature. First, this research is considered to be the first research that examines HIM from a holistic perspective using the science mapping technique. Second, it is envisioned that this research will guide future research and fill the gap in the field by providing valuable information about the common citation analysis themes and common keyword analysis themes in the field of HIM through bibliometric and document analysis.

This paper is divided into five sections. The first section provides general information about the subject and the second section outlines the research technique. The results of the analysis, performance analysis, bibliometric analysis, and prominent types of electronic records in the field of HIM are presented in the third section. The fourth section discusses the findings and limitations of the research, and the fifth section concludes the research.

## 2. Methodology and Data

### 2.1. Method

The bibliometric analysis method was used for science mapping in this study. Bibliometric analysis is considered a valuable tool for gaining an in depth understanding of large amounts of data, uncovering connections between publications in the field, exploring new avenues of research, and building a solid foundation in the field [31,32,33]. Especially in academic circles, such analyses are often used to evaluate article and journal performance, identify collaborative networks, and identify trends in the field [34,35,36].

This article uses the R Studio library Bibliometrix, which is free and open-source and was developed by Aria and Cuccurullo (2017) [37], as well as Python and Microsoft Excel. This software allows the direct import of bibliographic information from Scopus, Clarivate Analytics Web of Science, Dimensions, Lens.org, PubMed, or Cochrane Library to take place. It also allows one to create matrices to perform co-citation, merging, collaboration, and common word analysis. It can also provide a static picture of the field under study, with the option to divide the research period into different time periods when needed [37,38,39,40].

The research followed a scientific mapping process consisting of 5 stages, as proposed by Zupic and Čater (2015) [36]. These are (1) study design, (2) data collection, (3) data analysis, (4) data visualization, and (5) interpretation [37]. The study design and data collection followed the Preferred Reporting Items for Systematic Reviews and Meta-Analyses (PRISMA) principles, as shown in Figure 1.

### 2.2. Data

In this study, publications on “health information management” were searched in the WoSCC database on 25 October 2023. The WoSCC contains a large collection of bibliographic lists, citation networks, and full-text articles [34,35,36]. It is known as an important source for bibliometric and sub-metric analysis and is preferred by researchers due to its clear advantages. It is also a fundamental data source for bibliometric analysis and information mapping, covering a large part of the medical literature. It acts as a citation index and provides information on citations in publications. The WoSCC features, such as detailed records, data sets suitable for bibliometric analysis, and indexed prestigious publications, make it one of the most widely used sources for bibliometric analysis. The research data in the WoSCC database were obtained with these advantages in mind.

All documents written in English indexed in the WoSCC between 1983 and 2023 with the keyword “health information management” were searched (890 publications in total). The search was then narrowed to exclude publications that did not meet the inclusion criteria. This search included the “health information management” (subject) keyword in the topic; “Retraction” or “correction”, “Addendum” or “Biographical Article” or “Software Review” or “With-drawn Publication” or “Data Article” or “Meeting” or “Discussion” or “Bibliography” or “Reprint” or “Book” or “Correction” or “Note” or “Book Review” or “News Item” or “Letter” or “Book Chapters” or “Meeting Abstract” or “Editorial Material” or “Proceeding Paper” (excluding document types) (237 publications); and the exclusion of all publications “not in English (languages)” (23 publications). When only articles, reviews, and early-access publications were selected by publication type, 630 articles remained for bibliometric analysis.

### 2.3. Data Analysis and Visualizing

Using bibliometrics, it is possible to identify the key quantitative variables of a given research stream. It is also possible to follow the proposed five steps of the science mapping workflow. The data were analyzed using the R-based open-source software Bibliometrix [37,38,39,40]. First, a performance analysis of these articles was performed, including basic statistics, authors, number of publications, journals, institutions, and countries.

Data analysis requires descriptive analysis and network inference. Different approaches have been developed to extract networks using different units of analysis. In this study, co-occurrence and co-citation analyses were used. Co-occurrence analysis creates networks using keywords such as a common author, word, country, etc. For example, co-occurrence analysis uses the most important words or keywords in documents to examine the conceptual structure of a research area. It is the only method that uses the actual content of documents to create a measure of similarity. Paraphrase analysis produces semantic maps that facilitate an understanding of the cognitive structure of a domain. It can be applied to document keywords, abstracts, or full texts. The unit of analysis Is usually not a document, author, or journal, but a concept or keyword [37].

The co-citation link is established by the authors citing the documents under review. It examines the cited documents and helps to identify changes in paradigms and schools of thought when analyzed over time. The co-citation of two articles occurs when both are cited in a third article. Thus, co-citation is the counterpart of bibliographic coupling [37].

The Bibliometrix R-package allowed us to use the conceptual structure function to perform a multiple correspondence analysis (MCA) to draw a conceptual structure of the field and K-means clustering to identify clusters of documents that express common concepts. In this study, MCA was used for data reduction. “Biblioshiny” was also used as an application that provides a web interface for Bibliometrix to create a scientific map, including a co-citation network and co-word analysis. In the final stage, the most prominent types of electronic records were analyzed through a document analysis.

## 3. Findings

This section presents findings from the performance analyses, science mapping analyses, and analyses of the types of electronic records that are prominent in the field of HIM.

### 3.1. Performance Analysis

This section includes key information; the number of publications and citations by year; the top 10 most cited articles; the top journal, institution, country, and author rankings; and a trend topic analysis (Figure 2).

Figure 2 presents some of the main statistics of the scientific works published on “HIM” between 1983 and 2023. First of all, it can be seen that a total of 630 documents were published during this period, coming from 281 different sources (journals, books, etc.). This shows that the topic has a very large body of literature and has been covered in different publication channels. The annual growth rate of 11.78% indicates that there is continued interest and investment in this field and that the literature is growing rapidly. On average, each document received 11.11 citations, indicative of their high impact. The documents are relatively current, with an average age of 5.26 years, showing that research in the field is active. Regarding authorship and collaboration, the field commonly demonstrates multi-authored studies, as evidenced by only 42 single-author documents among the contributions of 2546 authors. On average, each document has 4.7 authors, and 23.49% of collaborations are international. Analyzing the document types reveals that there are the largest number of “articles”, at 541. There are also 44 documents in the “review” type. This finding shows that the literature is largely composed of original research. According to the results obtained, HISs are an active research area with a multidisciplinary approach, and high-impact studies are generally produced in this field.

The dataset records the total number of citations received from 1983 to 2023. The low number of citations between 1983 and 1993 suggests a lack of interest in the subject during that time. Citations show an irregular increase from 1994 to 2005, with 2006 and 2017 experiencing exceptionally high numbers. The recent decline in citations during 2022 and 2023 can be attributed to the large volume of new publications. Between 1983 and 1993, only one publication per year was recorded, suggesting minimal attention towards the research area during this time period. However, the number of publications has surged since 2012 and has reached its peak in the past two years, as depicted in Figure 1.

The studies listed in Table 1, which have been cited the most in the field of HIM, address several significant matters. The study by Mandel et al. (2016) [41] addresses the integration of medical applications across HISs through the Substitutable Medical Applications and Reusable Technologies (SMART) on the Fast Health Interoperability Resources (FHIR) platform. The objective of this research is to facilitate more effective communication between information systems and a more efficient management of patient information [41]. Kraemer et al. (2017) [42] examined the application of fog computing in the healthcare industry. They found that fog computing can perform a variety of tasks, from wireless devices to high network layers, and that it offers a balance between privacy and reliability concerns.

The “All Aspects of Health Literacy Scale (AAHLS)” [43] describes an instrument’s development and pilot that aims to measure health literacy in a multidimensional way. This scale provides critical dimensions for HIM, including the effective use of health information, effective communication with healthcare providers, and the management and evaluation of health information. Thus, the study aims to quantify the essential skills that are necessary for patients and healthcare professionals to make informed and impactful decisions [43].

The study by Kim et al. (2009) [44] focuses on the use and effectiveness of PHRs in a low-income and elderly population. In terms of HIM, the study highlights that PHR systems often fail to reach those that are the most in need but the least able to use them. Several factors such as limited access to technology, limited proficiency in computer and internet usage, an inadequate comprehension of health information, and restricted physical or cognitive abilities impair this population’s effective use of PHR systems. The study indicates that the use PHRs is restricted to only 13% of the population, and mostly during times when personal assistance is available. This implies that in order to create and implement effective HIM tools, socio-economic and age-related factors must be taken into consideration [44].

Lustria’s (2011) [45] study examines the digital divide in the use of eHealth technologies in the US and proposes solutions, such as web-based tools, to bridge this gap. Giakoumaki et al. (2006) [46] discuss the potential of digital watermarking technology in HIM to provide a solution for the safe and efficient storage and distribution of medical data. Ancker et al. (2015) [47] investigated how people with multiple chronic conditions handle their health information. This process encompasses several interactions between patients and providers and involves numerous tasks related to HIM, known as the “invisible work” [47]. Pratt et al. (2006) [48] explained how personal HIM helps individuals to take a more active role in their healthcare. Nouri et al. (2018) [49] provide criteria for assessing the quality of mobile health applications and discuss how these applications should follow a standard in terms of health services and information management.

The tenth most cited paper [50] examines the emergence of HIM in the household (HIMH) and its potential implications for consumer health informatics (CHI). The argument advanced in this paper proposes that a single individual is usually accountable for obtaining, handling, and arranging an assorted range of health information pertinent to HIM within the home. Paper-based tools continue to be extensively used, and households develop strategies to store information according to the urgency of their needs. This research highlights that the complex and robust strategies devised at home can effectively assist individuals in managing their personal health information, thereby contributing to consumer health innovations [50].

Overall, these studies address various dimensions of HIM, such as the role of technology, the need for standardization, personal and societal factors, and security and privacy. Technological approaches and methodologies are crucial in enhancing overall efficiency, security, and system accessibility. Nevertheless, these studies indicate that it is necessary to consider various cultural and social implications.

The leading journals, authors, universities, and countries are collectively displayed in Table 2, Table 3, Table 4 and Table 5 and Figure 3 based on performance metrics. As such, the *Health Information Management Journal* boasts the highest article count (134), while the *International Journal of Medical Informatics* and the *Journal of Medical Systems* demonstrate the lowest article count (eight). Kim S was the most productive author, publishing 12 articles, while Casper G and Freitas A had the lowest productivity, each publishing only four articles. The University of Washington Seattle appears to be the most productive institution with 21 articles, whereas the Florey Institute of Neuroscience and Mental Health and Monash University had the lowest productivity, each publishing only 13 articles. The United States had the highest number of publications, totaling 199, while the Netherlands had the lowest number with a total of only 22 (Table 5).

The topic trend graph (Figure 2) displays the frequency of utilization for different keywords in the realm of HIM and their popularity as time progresses. The aim of this analysis is to furnish researchers with an overview of the topics that are gaining significance in the field, those which uphold consistent interest, and those that belong to certain time periods. “Health Information Management” is the most frequently used keyword for popular topics due to its topicality, with a total of 326 usages. The prominence of this term has increased from 2017 to 2022, showcasing the topicality of the subject. The impact of the COVID-19 pandemic on the research field is clearly evident. In fact, the keyword “COVID-19” appeared 18 times in the last two years (2021–2022), highlighting its significant impact on HIM. In terms of recent developments, the phrases “Clinical Coding” and “Health Information Systems” have become increasingly popular, particularly in 2020 and 2021. This study reveals an ascending trajectory in the domains of health information systems and clinical coding. The terms “Electronic Health Records” and “Medical Informatics” demonstrate enduring prominence over an extensive time frame (2016–2022 and 2014–2022), suggesting a consistent significance and relevance in these fields. Less commonly used and specific terms include “Nurses”, “Self-Care”, and “Africa”, which had a low frequency (four) and were popular only during a specific time period (2012–2015). This discovery implies that these topics are associated with a particular time period or are less researched. The phrase “Diagnosis-Related Groups” was only prevalent in 2017, indicating the existence of topics that are time-limited. The frequency and time distribution of keywords indicates which topics are gaining momentum, enjoying stable attention, or are period-specific. This type of trend analysis offers important insights for researchers and policymakers to identify relevant topics to focus on (see Figure 2).

Upon analyzing Figure 3 using a word cloud analysis, it is evident that the most utilized and significant keywords in the dataset are “health”, “information”, and “management”. Additionally, the terms “data”, “system”, and “clinical” feature prominently within the dataset. The high frequency of “electronic”, “medical”, and “systems” suggests their importance to HISs and electronic health records. Additionally, the prevalence of “care”, “hospital”, and “patients” demonstrates the direct link between HIM and patient care and hospital management. The use of terms like “quality”, “research”, and “technology” emphasizes the need to integrate HIM with quality improvement, research, and technological advancements. Similarly, the inclusion of terms like “Ghana”, “national”, and “public” underscores the importance of considering HIM in relation to local and national health policies and public health. This word cloud highlights critical concerns, current tendencies, and possible research gaps deserving the attention of HIM specialists and researchers. The terms used also demonstrate the extensive scope of HIM and its interconnectedness to multiple healthcare components (Figure 3).

### 3.2. Multiple Correspondence Analysis

The data were analyzed using MCA and each term was positioned on two dimensions (Figure 4). These dimensions represent the relationships and similarities between the terms. Furthermore, all terms were located in the same cluster (Cluster 1). This shows that they are grouped around a common theme.

The rules for interpreting the MCA graphs are as follows: axes represent variance, the position of terms indicates similarity, axis values indicate characteristics, clustering indicates similar themes, end points indicate special themes, symmetry indicates a balanced relationship, and the interpretation of dimensions depends on context. It is important that the graph is clear and interactive [37].

Terms with high values in Dim.1 and Dim.2 may indicate the future focus of the field. For example, high values for terms such as “blockchain”, “artificial intelligence”, and “data accuracy” may indicate that these technologies and concepts will become increasingly important in the field of HIM.

Technological developments: Terms such as “blockchain”, “artificial intelligence”, and “internet of things” suggest that technological innovation will play an important role in HIM. This may signal an increase in technological solutions to issues such as health data security, data analytics, and improving the quality of patient care.

International and coding standards: The prominence of terms such as “International Classification of Diseases” and “ICD-10” suggests that international health coding standards and classifications will continue to be important in HIM. This may be important for the development and implementation of global health data standards.

Patient safety and data quality: Terms such as “patient safety” and “data quality” suggest that patient safety and data quality will remain central elements of HIM. This may indicate that healthcare organizations will invest more in these areas and shape their policies accordingly.

Education and research: The terms “education” and “research” emphasize the importance of continued development and innovation in these areas. Education and research in HIM may be critical to keeping pace with new developments in the sector.

Dim.1 (X axis): This axis explains 36.81% of the variance in the dataset and may represent the overall theme or weight of publications in HIM. Terms that move to the right in Dim.1 may indicate a more central position within the field or represent more recent and advanced topics. Dim.2 (Y axis): This axis explains 15.89% of the variance in the dataset and may represent another important feature or relationship of the terms. It may suggest that terms that move upwards in Dim.2 are perhaps more closely related to specific sub-areas or specialized topics within HIM. In conclusion, this MCA graph can be used as an important analytical tool to understand the current situation and possible future directions in the field of HIM. The interpretation of this graph illustrates the broad and diverse nature of the field and important development trends.

### 3.3. Scientometric Analysis

In this section, we conducted co-citation and co-occurrence analyses to determine the prominent color clusters. The names of each color were determined by utilizing the prominent color clusters. Additionally, the intellectual and conceptual structures, along with the electronic record types, of publications on HIM were determined.

#### 3.3.1. Co-Occurrence Network Analysis

When analyzing the co-occurrence analysis of HIM, three distinct clusters are revealed (Figure 4). Theme suggestions for each cluster and corresponding clarifying vocabulary are provided below.

*Theme 1: Health information management (red cluster):* This theme concentrates on health information and data management, storage, and analysis. The keywords for this theme comprise “Health Information Management”, “Electronic Health Records”, “Medical Informatics”, “E-health”, and “Education”. This theme comprises five sub-themes explained below.

-*Sub-theme 1: Electronic records (ERs):* This sub-theme analyzes digital record systems such as EHRs and EMRs. ER systems facilitate the storage and easy sharing of patient information in a digital environment.-*Sub-theme 2: Medical informatics:* This sub-theme focuses on the technological tools and methods used to collect, store, and analyze health information. The use of medical informatics is essential for ensuring prompt and precise decision making, particularly in the diagnosis and treatment of diseases.-*Sub-theme 3: E-health and telemedicine:* The use of digital platforms and remote access tools for delivering healthcare services is referred to as e-health and telemedicine. The importance of these services is amplified during extraordinary circumstances, such as pandemics.-*Sub-theme 4: Health education and awareness:* Health education is a crucial element in enhancing healthcare quality and guaranteeing patient safety. This sub-theme highlights the significance of education for healthcare providers and the general population.-*Sub-theme 5: Health information systems:* This sub-theme addresses HISs, which are crucial for the functioning of healthcare organizations. These systems manage a variety of transactions, including patient records, laboratory results, billing, and treatment plans.

The red cluster addresses diverse aspects and priorities of overall HIM. The theme identifies multiple subfields and specific challenges within the healthcare industry, paving the way for further in-depth investigation and potential solutions. The five sub-themes signify various facets and subjects in the HIM field, each with a precise focus and unique problems to be resolved.

*Theme 2: Personal health information (blue cluster):* This theme explores how individuals manage their health information, with emphasis on “Information Management” and “Personal Health Information Management”. The theme raises concern about how PHRs can help individuals store, share, and manage their personal health information. This is critical for individuals to effectively monitor their health and communicate with healthcare providers.

*Theme 3: Clinical coding and data quality (green cluster):* This theme focuses on data management and coding practices in the healthcare sector, specifically involving clinical coding, data quality, medical records, hospital information systems, ICD-10, data accuracy, and primary health care. The theme specifically addresses technical aspects, including clinical coding, hospital information systems, and international classifications of diseases. Additionally, ensuring data quality and accuracy is crucial for providing improved patient services and ensuring patient safety.

#### 3.3.2. Co-Citation Analysis

Co-citation analysis is a bibliometric technique for identifying connections among studies and themes within a given research field. In this investigation, an examination of co-citation analysis shows that all studies are grouped into six distinct clusters (Figure 5).

The red cluster is proposed to be named “HIM and Technology Adoption”, which investigates the optimal methods and timing for implementing HISs and technology. The implementation of HIM has the potential to enhance patient safety, treatment procedures, and communication among healthcare providers. Successful technology adoption is an integral part of this process. Highlighted papers offer essential insights into how these systems and technology can improve the quality and efficiency of healthcare.

HIM plays a critical role in healthcare delivery and quality indicators. A well-designed and executed system can improve treatment procedures and facilitate communication among healthcare providers [51,52,53]. HIM plays an active role in delivering, managing, and planning healthcare services. Various technological tools are integral components of these systems, such as personal health records and patient portals. Understanding technology acceptance is crucial when considering the adoption of these tools [54,55].

Tang et al. (2006) [54] discussed the definition and benefits of personal health records and strategies for addressing barriers to their adoption. Likewise, Detmer and colleagues (2008) [56] contend that such registries serve as revolutionary resources in consumer-centered healthcare [56]. Irizarry et al. (2015) [57] provide a comprehensive evaluation of the advantages offered by health information technologies and how they revolutionize the interaction between healthcare providers and patients. Technology acceptance is critical in HIM. The effective adoption and use of technology by individuals and healthcare providers will be decisive in improving the quality and efficiency of healthcare. The articles in this cluster illustrate how HIM and technology acceptance play a critical role in this process.

Regarding the blue cluster, “Australian Health Services and Clinical Coding” addresses the country’s health status and the quality and safety of health services. The cluster includes the Australian Institute of Health and Welfare’s research on chronic disease mortality and health status [58,59], as well as the Australian Health and Quality Commission’s agenda to improve quality and safety [60].

HIM serves as a crucial instrument for efficiently planning, managing, and delivering healthcare services. An HIS collects, processes, and analyzes data that can help healthcare providers and policymakers develop strategies to improve quality and efficiency. Furthermore, the publications in this cluster accentuate the specialized domains of HIM, encompassing the ICD and clinical coding. These coding systems are critical for developing strategies for the management and prevention of specific diseases and for allocating health resources more effectively.

The green cluster, “Health Information Systems Success”, addresses methodological and applied issues for assessing the success and effectiveness of these systems. This includes modeling information systems’ success [61] and predicting the use of individual health records [62]. DeLone and McLean (2003) [61] propose a model for measuring the triumph of information systems and scrutinize its potential effects on the quality and effectiveness of healthcare. Similarly, Ford, Hesse, and Huerta (2016) [62] examine how predicting future usage levels of individual health records can influence the consumer-driven nature and success of HISs. The theme “Health Information Systems Success and Evaluation” offers methodological and practical perspectives on how we can evaluate and improve the effectiveness and success of these systems. The work of Fornell and Larcker (1981) [63], often referred to in the methodology section, provides a sound basis for these evaluations. The articles in the purple cluster comprehensively address the impact of EHRs and eHealth technologies on the quality, efficiency, and cost of healthcare. Concepts such as “Meaningful Use” are critical to increase the effectiveness and utility of these technologies [28]. Furthermore, systematic analyses suggest that these technologies can significantly improve healthcare’s quality and efficiency [25,26]. The deployment of EHRs and eHealth technologies can improve health service performance in areas such as time efficiency, quality, and cost saving [27,64,65]. These findings highlight the significance of integrating and utilizing these technologies to enhance healthcare service quality and efficiency. The purple cluster offers a comprehensive examination of how HISs can efficiently incorporate and employ these technologies, and how they can revolutionize healthcare.

The theme of “Personal Health Information Management (PHIM) and Consumer Health Informatics (CHI)” is the focal point of the orange cluster. Many authors and articles have extensively covered this theme. For instance, Moen and Brennan (2005) [50], Civan et al. (2006) [66], Pratt et al. (2006) [48], and Unruh and Pratt (2008) [67] offer significant perspectives on how to self-manage personal health information and communicate it with healthcare providers. These authors also discuss the influence of PHIM on CHI and its potential for improvement. Piras and Zanutto (2010) [68], Valdez et al. (2015) [9], and Or and Karsh (2009) [69] have studied CHI, discussing how it can enable individuals to better access health services and improve their overall quality of healthcare. Zayas-Cabán’s (2012) [70] study on home HIM offers valuable insights into how individuals can manage their health information.

Electronic personal health records (ePHRs) have been evaluated by several authors, including Ancker et al. (2015) [47], Agarwal and Khuntia (2009) [71], Archer et al. (2011) [72], and Halamka et al. (2008) [73]. The authors assessed the advantages and disadvantages of ePHRs for both users and healthcare providers. Kim et al. (2009) [44] addressed challenges related to the utilization of ePHRs among low-income elderly populations, and Holden et al. (2013) [29] explored the implementation and customization of bar-coded medication administration technology in healthcare. Finally, Mickelson, Willis, and Holden (2015) [74] examined medication-related cognitive formations and provided valuable data on how older adults manage medications for heart failure. These articles offer crucial insights into how personal HIM and consumer health informatics are utilized in various settings and how to address challenges and opportunities in these fields.

The brown cluster is titled “Country Health Information Systems Framework and Standards”. HISs are essential for the planning, monitoring, and evaluation of health services. The World Health Organization (WHO) offers guidance on HIS improvement and structuring in the report by presenting a framework and a set of standards for country HISs (the WHO, 2008) [11]. The study conducted by the WHO offers a comprehensive overview of how HISs can be utilized by countries to enhance health services and improve public health outcomes.

## 4. Discussion

The growing volume and scope of research conducted in the field of HIM in recent years have improved our comprehension of the sector’s dynamics and the critical significance of HIM. This research encompasses several complex and diverse themes, including patient data management, the adoption of new technologies, quality control, safety concerns, and governance and policy regulations.

Recent research has revealed that personalized healthcare is increasingly prevalent [75,76]. Therefore, it is crucial to re-evaluate management procedures for processing patient-generated data of a high quality. Additionally, the growing concern of data breaches in healthcare facilities has led to the necessity of establishing secure data management systems [77]. The significance of technology in this field cannot be overstated. Emerging technologies like blockchain and Health Level Seven (HL7) FHIR demonstrate the need for careful consideration regarding interoperability and data integrity [78,79].

Digital tools, such as electronic health records and patient information systems, have a vital role in enabling healthcare professionals to make well-informed treatment decisions [80]. Multiple studies suggest that HIM needs to be continuously improved, particularly in terms of safety and quality [5,81,82]. Governance and policy making play a crucial role in healthcare management, as emphasized by Barbazza et al. (2022) [83] and Begany and Martin (2020) [84].

Over the past few years, research has enhanced our comprehension of the significance of HISs in the healthcare industry. Specifically, examining specialized topics like environmental health [85], critical incidents and emergencies [86], and engineering strategies for complex systems [87] highlights the vital role of HIM in healthcare administration. The existing literature highlights the significance and intricacy of HIM while revealing diverse benefits and obstacles that these systems entail in the healthcare industry. Bringing the research together is essential for comprehensively understanding the sector’s dynamics and the role of HIM while establishing a strong foundation for future developments.

### 4.1. Evaluation in Terms of Performance Analysis

In discussing the results of the performance analysis, as well as the publication and citation analyses, the most cited studies, the journal, author, institution, and country rankings, the keyword and trend analyses, and the factor analysis results were evaluated.

The distribution of publications and citations by year shows the growth and development of the field of HIM. While the low number of citations between 1983 and 1993 indicates that interest in the subject was limited during this period, the citation peaks in 2006 and 2017 indicate the great impact of the studies carried out in these years on the field. The increase in the number of publications, especially in the last decade, shows the dynamism and growing popularity of the field.

The analysis of the most cited studies reflects the main themes in HIM and the diversity of the field. Studies such as those by Mandel et al. (2016) [41] and Kraemer et al. (2017) [42] highlight the importance of technological integration and innovation in HISs. On the other hand, studies such as those by Kim et al. (2009) [44] and Lustria (2011) [45] draw attention to how HISs are affected by socio-economic and demographic factors. These studies show that HIM is both technologically and socially complex.

The ranking of the most productive journals, authors, institutions, and countries in terms of HIM publications shows the geographical and academic contexts in which this field is more intensively studied. In particular, the high productivity of the USA and the University of Washington, Seattle, indicates that these geographical and institutional contexts make important contributions to HIM.

The keyword and trend analyses show the prominent themes in HIM and how these themes have changed over time. The frequent use of terms such as “HIM”, “COVID-19”, “clinical coding”, and “electronic health records” underlines both the current and long-term importance of these issues. These analyses can help researchers and policymakers to identify areas where they should focus.

The MCA results show the relationships and similarities of terms in the field of HIM. Terms such as “blockchain”, “artificial intelligence”, and “data accuracy" have high scores, indicating the future importance of these technologies and concepts. This analysis reveals the technological and conceptual evolution of the field of HIM.

This discussion illustrates the breadth and diversity of publications in the field of HIM. Several themes such as technological innovation, socio-economic and demographic factors, international standards, patient safety, and data quality emerge as key elements that will shape the future of the field. In addition, these analyses emphasize that the field of HIM is a constantly changing and evolving one, and that researchers should direct their studies with this dynamic in mind.

### 4.2. Evaluation in Terms of Co-Citation Analysis

An important issue is how the themes that the co-citation analyses reveal about HIM are similar to recent studies or whether they are in line with older themes.

The themes of HISs and technology adoption frequently centers on the effective management strategies and adoption of novel technologies. Recent studies highlight the rapid acceleration of technology adoption, necessitating the revision of management strategies [75,76]. Previous themes may relate to broader work on technology adoption. In contrast, the clinical coding and quality improvement themes concentrate on issues of quality and safety. New studies can inform efforts to improve quality by analyzing various geographies and management systems [5,81,82]. Past research often focused on broader themes of quality and safety, whereas contemporary studies provide more distinct and applicable recommendations.

The theme of HIS success and evaluation emphasizes the continuous improvement of HIM. Recent studies provide more detailed guidelines on how to evaluate systems and measure their success [80]. Preceding themes typically examine what constitutes HIM success and how it can be measured within a wider theoretical context.

The theme of electronic health records and the impact of eHealth technologies centers on data integrity and how it can improve treatment outcomes. Recent in-depth research has focused on data integrity and analytics [78,79]. The personal HIM and consumer health informatics theme emphasizes the significance of personalized healthcare and effective data management. A recent study focuses on how to effectively manage patient-generated data and personalized treatments [77].

As a result, co-citation analyses and recent research may align with previous themes or offer more precise and practical instructions, or they may differ. Understanding this harmony and discrepancy can provide a thorough analysis of health service management.

### 4.3. Evaluation of Co-Occurrence Analysis

Comparing themes that arise from co-occurrence analyses with newly emerging concepts can offer valuable insights into the current state of the literature.

Recent research on HIM has uncovered its growing depth and breadth within the healthcare industry. These studies offer insight into the essential role played by HIM. Specifically, recent HIM research delves into multifaceted themes, including patient data management, technology adoption, quality and safety concerns, and policy and administrative arrangements. Recent studies indicate that HIM is connected to an escalating pattern of individualized healthcare. This trend necessitates a re-evaluation of management tactics for ensuring the quality processing of patient-generated data. Additionally, the security and data handling of HIMs requires the resolution of data breaches and security issues. New technologies, specifically blockchain [21,82] and HL7 FHIR [79], signal the need for careful consideration of HIMs with regard to factors such as interoperability and data integrity.

More recent studies have placed greater emphasis on technical issues such as clinical coding, hospital information systems, and data quality. These issues are crucial for data management and maintaining high-quality healthcare information systems. Accurate data are closely linked to both data breaches and the overall quality of healthcare.

### 4.4. Limitations

This study highlights the potency of bibliometric and citation analyses in assessing research scope and impact. However, it is crucial to note that these analyses do not provide direct information about research quality. The limitations of these analyses have propelled interest in scientific metrics [32,88,89]. It is important to recognize that only the WoSCC database was used in this study and other databases were not considered. The search was limited to keywords. Among the publications found, only English articles were analyzed. Publications in other languages with undoubtedly valuable findings may have been overlooked. Despite these limitations, the findings could potentially be a motivation for future research [90,91].

### 4.5. Future Research Agenda

The research questions are based on two main sources: the questions prepared by utilizing the findings and discussion part of this article and the studies focusing on future research questions specifically in the field of HIM constitute two mirrored sources. The future research topics and questions were prepared under a total of 10 headings.

When developing future research questions in the field of HIM, it is essential to consider the findings and trends in the existing literature. Topics such as blockchain technology, big data analytics, and patient-centered healthcare systems are at the forefront of innovation and research in HIM. In this context, the study by Wenhua et al. (2023) [92] provides a comprehensive perspective on the healthcare applications, security issues, challenges, and future trends of blockchain technology. This study is a fundamental resource for an in-depth understanding of the implications and applications of blockchain technology in HIM research.

Ahmed et al. (2023) [93] provided a comprehensive review of how big data analytics can be used and optimized in the healthcare industry. This study outlines the frameworks, applications, and implications of big data use in the HIM field, providing an important foundation for future research. The study conducted by Aldamaeen et al. (2023) [94] presents the key requirements and framework for personal health records based on blockchain technology. This study provides an excellent reference for enhancing patient-centered approaches and strengthening technological infrastructures in the HIM field. Finally, O’Dell and Gabriele (2023) [95] work towards the development of a comprehensive personal health record (PHR) to improve the healthcare experience. This study provides important insights into how PHR systems can be developed to improve patient engagement and health management practices, and provides a methodological framework for future research.

These articles cover current technology trends and management strategies that are critical to future HIM research (Table 6). Therefore, it is imperative to utilize these resources when developing promising HIM research questions in order to generate real-world solutions that are grounded in the current scientific literature and focused on practice. Each resource provides HIM researchers with insights into the future of healthcare, policy, and technology use, enabling the development of innovative and effective solutions that will contribute to the advancement of the field.

## 5. Conclusions

This study provides a comprehensive analysis of HIM publications between 1983 and 2023. The results show the rapid growth and development of the field. The publication of 630 documents and an annual growth rate of 11.78% shows that the field of HIM has a large body of literature and is actively addressed in different publication channels. On average, each document received 11.11 citations, indicating that the research has a high impact. In addition, the average age of the documents is 5.26 years, indicating that the field is constantly being updated and that research is active.

The most cited studies highlight the diversity and scope of the HIM field. These studies cover a wide range of topics such as the integration of HISs, the implementation of fog cloud computing, the measurement of health literacy, the use and effectiveness of PHRs in low-income and elderly populations, methods for bridging the digital divide, and the potential of digital watermarking technology in HIM. These studies show that the field of HIM is not limited to technological innovations, but also includes socio-economic, demographic, and cultural factors.

Finally, this research provides valuable insights into future trends in HIM. Technological advances and methodological innovations increase efficiency and safety while improving accessibility to systems. However, studies in this field need to consider not only technological aspects, but also personal, societal, ethical, and political dimensions. HIM, with its multidisciplinary approach, has an important role to play in improving the quality of healthcare and ensuring patient safety. Therefore, it is important that research and development in this field is approached from a broad perspective and is continuously updated.

## Data Availability

Publicly available datasets were analyzed in this study. This data can be found here: https://www.webofscience.com/wos/woscc/summary/03d155c8-76eb-464c-a566-58caa47111b9-af3f2da4/times-cited-descending/1 (accessed on 25 October 2023).

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
