# Peer review of "Analysis of Publications on Health Information Management Using the Science Mapping Method: A Holistic Perspective"

_healthcare, 2024, doi:10.3390/healthcare12030287_

Round 1

Reviewer 1 Report

Comments and Suggestions for Authors

The file is in the attachment

Author Response

Dear Reviewer,
Thank you for taking the time to read and review our article. Your valuable comments and contributions to our article were very important to us. Undoubtedly, your evaluations and suggestions helped us to make the article more effective and valuable. The file showing the changes made in the article titled "Analysis of publications on health information management using the science mapping method: A holistic perspective" is attached.
Corresponding Author

Reviewer 2 Report

Comments and Suggestions for Authors

The affiliation details should conform to the journal requirements.

Abstract: Authors should indicate what the abbreviation "HIM" refers to.

Figure 1: Words were shown incompletely.

Figure 1 and main text were contradictory: Figure 1 showed that the authors searched for "Health Information Management accreditation", but main text mentioned ""Health Information Management". Similarly, Figure 1 showed that Retraction, Correction etc were searched for, but main text mentioned only articles and reviews were included. Please rectify.

Results: "...addresses the integration of medical applications across health information 206 systems through the SMART on FHIR platform". Again, please spell out SMART and FHIR in full for their first appearance.

Methodological details of co-citation network and co-word analysis should be described in the Methods section. Figures 4 & 5: Any thresholds to determine whether a node should be visualized or excluded? How to determine how many clusters should be used?

Figure 5: Some institution names were listed instead of [Author, Year]. Please explain.

Discussion: Results from the "basic performance analysis" were not discussed.

Limitations:"... only English-language articles were analyzed in WoS, which may not provide a full reflection of global research, as pre-1975 articles are not accessible." But the search method only searched for 1983-2023. So this limitation point was not relevant?

Annex 1: It was truncated.

Author Response

(The authors gave the same response as above.)

Reviewer 3 Report

Comments and Suggestions for Authors

Perihan Senel Tekin et al. submitted an interesting bibliometric analysis about the medical publications. The topic was of a certain significance, and could be reconsidered after a Major Revision. Detailed comments:

1.       The format of author affiliations was probably incorrect. Please provide the full information.

2.       In the Introduction, the reason why HIM could be divided into three periods (what were the criteria) must be demonstrated.

3.       Figure 1 should be redrawn following the PRISMA principles. Besides, the information in this figure was not completely shown.

4.       In the paragraph below Figure 1, the authors stated that they consulted WOS database; but in Figure 1, it was WOSCC. Please use unified expressions.

5.       References should be cited in Table 1.

6.       In Section 3.2.1., the strength values could be supplemented along with the theme analysis.

7.       Main findings in this work must be summarized in the Conclusion Section.

8.       Annex 1 seemed not to fully display.

9.       Please double-check the format of References.

Author Response

(The authors gave the same response as above.)

Round 2

Reviewer 1 Report

Comments and Suggestions for Authors

The authors have already revised all of the reviewer's issues. Thank you for carefully revising point-by-point and improving your research. I have no objection. The manuscript is suitable for publication in the Healthcare.

Author Response

Dear Reviewer,
Thank you for your valuable time and contribution to our article.
Best regards,
Corresponding Author

Reviewer 2 Report

Comments and Suggestions for Authors

Why are there so many questions listed at the end of the discussion before the Conclusion? It looks so weird. If these questions are for future perspectives for fellow researchers, please put them into a figure or table, not in the main text. 

Author Response

Dear Reviewer,
Thank you for your valuable time and contribution to our article. Your comments and suggestions were very valuable to us. With this in mind, we have added a table (Table 6) to the "Future Research Agenda" section of our article. You can see the revised version in the attached file. Thank you again for your support and contribution. Best regards,
Corresponding Author

Reviewer 3 Report

Comments and Suggestions for Authors

Thanks for your revision.

Author Response

Dear Reviewer,
Thank you for your valuable time and contribution to our article. Your comments and suggestions were very valuable to us. Thank you again for your support and contribution.

Best regards,
Corresponding Author